# Low Compaction Level Detection of Newly Constructed Asphalt Pavement Based on Regional Index

**DOI:** 10.3390/s22207980

**Published:** 2022-10-19

**Authors:** Jiaming Tang, Zhiyong Huang, Weixiong Li, Huayang Yu

**Affiliations:** 1Xiaoning Institute of Roadway Engineering, Guangzhou 510000, China; 2School of Civil Engineering and Transportation, South China University of Technology, Guangzhou 510000, China

**Keywords:** three-dimensional ground-penetrating radar, low compaction level detection, regional index, dielectric constant, newly constructed asphalt pavement

## Abstract

In order to improve the prediction accuracy regarding low compaction level of asphalt pavement, this paper carries out indoor tests to detect the voids and dielectric constants of AC-13, AC-16 and AC-25 asphalt mixtures, obtaining their relationship equations via linear fitting and determining the dielectric constant judgment threshold of low compaction level segregation risk points $\varepsilon_{1}$. Based on the common mid-point method, three-dimensional ground-penetrating radar is used to obtain the dielectric constant of the physical engineering test section. The researcher can draw the distribution map of the low compaction level segregation risk area according to the judgment threshold $\varepsilon_{1}$ of the rough segregation risk points; divide the connected risk areas; determine the regional convex hull; and calculate the regional indicators such as the regional area, the ratio of the convex risk points and the mean value of the regional dielectric constant. The response surface analysis method is used to acquire the model of risk area index and core void ratio. The model is employed to predict and verify the core void ratio in the risk area of the road section and verify the accuracy of the model. The results show that the error range between the predicted voids and the measured voids is −0.4%~+0.4%, and the mean absolute value of the error is 0.25%. Compared with the mean measured voids of 6.63%, the relative error is 3.77%, indicating that the model can accurately predict the regional low compaction level segregation degree.

## 1. Introduction

One important reason for damage to asphalt pavement in its initial stage is the segregation of the newly constructed asphalt pavement. The segregation of asphalt pavement can be divided into compaction level segregation, temperature segregation and thickness segregation [1]. Low compaction level segregation belongs to the first of the three segregation types mentioned before. During the construction of asphalt pavement, large-size aggregate is locally concentrated, leading to the low content level of the asphalt mortar and fine aggregate, hence low compaction level segregation in the sense that the asphalt mortar cannot fully fill the voids between the coarse aggregates due to the voids being too large, resulting in poor anti-fatigue and anti-tensile properties of the pavement. The coarse aggregates can be easily peeled off, causing early water damage and forming pits [2]. Low compaction level segregation of asphalt pavement is mainly related to the quality fluctuation of mixture, rolling temperature, paver performance and other factors during construction. Strengthening the detection of low compaction level segregation of asphalt pavement, timely taking effective measures to treat the segregation area, analyzing the causes of low compaction level segregation and adjusting the paving construction process are of great significance to reduce the early damage of asphalt pavement and prolong the service life of asphalt pavement [3].

At present, destructive methods such as coring are often used to detect and evaluate the low compaction level segregation areas of new pavement. The location of coring disturbs the original pavement structure, and the coring repair materials often use solidified materials with good fluidity, such as cement paste and polyurethane materials, which are quite different from asphalt pavement materials. Due to sudden stiffness and stress concentration, the repair interface is the weak point of early damage. Therefore, coring seriously damages the integrity and tightness of the asphalt pavement. Even though the core filling and other treatment measures are taken, the core filling effect is poor, and it is often difficult to achieve good results, resulting in a series of early malfunctions of the new pavement. In addition, the traditional core drilling method is time-consuming and laborious. It also demonstrates poor accuracy, low degree of automation, a small sample size, strong subjectivity and large error [4,5]. 

Chen et al. [6] scanned the micro-texture features of asphalt pavement through laser texture detection technology so as to detect the surface segregation of asphalt pavement. Xiong [7] used the PaveTrack TM Plus (PQI380) to evaluate the density of asphalt pavement and detect the aggregate segregation area with low density. To some extent, these methods can predict the distribution of asphalt pavement segregation area, but they cannot use vehicle onboard detection and detect the segregation quickly in a large area. Wang et al. [8] adopted infrared temperature measurement technology to quickly detect and evaluate the pavement permeability in high-temperature weather based on the temperature difference of specimens with different voids under the same illumination time. The infrared measurement detects the permeability coefficient, which is related to the compaction level to a certain extent, but it is not a direct indicator for evaluating the compaction level. Moreover, at high speed, the temperature detection accuracy of the infrared measurement is about 1 °C. The temperature difference of asphalt pavement in areas with different permeability coefficients is relatively small, ranging from 0.9 to 5.7 °C. Therefore, the infrared temperature measurement can only achieve the permeability coefficient gradation.

Researchers use ground-penetrating radar (GPR) to detect the low compaction level segregation area of asphalt pavement quickly and non-destructively. Luo [9] et al. used three-dimensional ground-penetrating radar to detect the dielectric constant of asphalt pavement and built a prediction model for the relationship between the dielectric constant of asphalt pavement and the porosity detected by PQI380 based on a multi-layer feedforward neural network. Based on an improved extended common mid-point method, Zhen [10] used ground-penetrating radar to quickly detect the dielectric constant of asphalt pavement and verified through the test field that the model was significantly better than the reflection coefficient method in detecting the dielectric constant of thick asphalt layer (thickness > 10 cm). Shao et al. [11] analyzed the GPR signal of asphalt pavement according to the non-linear optimization method of gradient descent to estimate its thickness and density, with errors of 3 mm and 1.81%, respectively. As a fast and non-destructive detection method, GPR also shows good accuracy in the prediction of asphalt pavement segregation.

At present, the final prediction results proposed by the rapid and non-destructive testing technology for low compaction level segregation of asphalt pavement are still evaluated separately according to the single point test results, which are completely independent of each other without considering the correlation between adjacent points. However, during the construction of asphalt pavement, the distribution of construction quality often presents a certain regionality, that the aggregate gradation, paving temperature, compaction temperature, compaction power, etc., between two close points are relatively similar (normally within 1 m). When analyzing the construction quality of asphalt pavement, the overall distribution trend and law of pavement segregation shall be considered from the regional perspective. The low compaction level segregation area of asphalt pavement is predicted based on the regional related indicators. On the one hand, it can eliminate the influence of outliers on the detection results and improve the prediction accuracy to a certain extent. On the other hand, the output of regional test results can more clearly reflect the overall distribution trend of low compaction level segregation on asphalt pavement. It also has certain advantages over the output of single point results in the treatment of low compaction level segregation areas and the optimization of construction technology based on the test results.

In order to solve the problem of single point prediction, this paper predicts the low compaction level segregation area of asphalt pavement from the regional index with the help of the relevant theories and research results of 3D ground-penetrating radar, dielectric constant detection and digital image. 

## 2. Dielectric Constant Detection Principle of 3D Ground-Penetrating Radar

In this paper, the Geoscope three-dimensional ground-penetrating radar system from 3D-Radar was adopted for research. The system mainly consists of a host computer, real-time dynamic positioning system and a multi-channel antenna array, as shown in Figure 1a. The multi-channel antenna array of three-dimensional ground-penetrating radar has a scanning width of 1.5 m each time. By setting multiple detection channels, the full cross-section coverage scanning of any road can be realized. During detection, the trigger distance parameter was set as 0.03 m, the depth range was set as 62.5 ns and the dwell time was set as 1 ms. 

There are three main methods for the dielectric constant of asphalt pavement by GPR: reflection coefficient method, thickness constant method and common mid-point method [12]. The reflection coefficient method does not consider electromagnetic wave diffraction and can only realize point measurement [13]. The premise assumption of the constant thickness method is the asphalt pavement thickness being a constant value, which has a large error with the actual situation of the project [14]. The common mid-point method is one of the dielectric constant detection methods, which is able to be carried out in a large area, with well-documented theoretical calculation model and accurate results [15].

In this paper, three-dimensional ground-penetrating radar is used to detect the dielectric constant by the common mid-point method through different antenna combinations [16] so that the two-way travel time of the reflected signal at the interface of the structural layer is different, and the electromagnetic propagation equations containing the dielectric constant are established to solve the dielectric constant of the pavement structural layer. As shown in Figure 1c, a total of nine antenna combinations with the same mid-point but unequal spacing were set during the detection.

As shown in Figure 1b, different antenna combinations are formed by adjusting the spacing between the antenna transmitting unit and the receiving unit. The antenna combination T1/R1 is an adjacent antenna group, and the antenna combination T2/R2 is an antenna group with a spacing of meters. Two electromagnetic wave propagation equations can be obtained for antenna combinations with different spacing. According to the propagation law of electromagnetic wave, the T1/R1 antenna combination fits:(1)$$2d_{1}=\frac{c}{\varepsilon_{r,1}}t_{1}$$

In Equation (1), $d_{1}$ is the thickness of the structural layer detected, $\varepsilon_{r,1}$ is the relative dielectric constant of the asphalt layer, c is the speed of light and $t_{1}$ is the two-way travel time, which can be obtained from the radar image by tracking the interface track, as shown in Figure 2a.

The T2/R2 antenna combination fits:(2)$$2\sqrt{d_{1}^{2}+{\left(\frac{x}{2}\right)}^{2}}=\frac{c}{\varepsilon_{r,1}}t_{2}$$

In Equation (2), x is the distance between the transmitting antenna and the receiving antenna. $t_{2}$ is the double-layer travel time able to be obtained from the radar image by tracking the interface track, as shown in Figure 2b. By combining simultaneous Equations (1) and (2) to offset parameter d1, the following can be obtained:(3)$$\varepsilon_{r1}=\frac{c^{2}}{x^{2}}(t_{2}^{2}-t_{1}^{2})$$

In Equation (3), $t_{1}$ and $t_{2}$ can be obtained from the radar image, and x is known: it is the distance between the receivers that were selected. In this study, x is 0.15 m, while c is the speed of light, a known factor. The dielectric constant of the asphalt structure layer at all measuring points in the scan area can be obtained through Equation (3).

## 3. Indoor Test

### 3.1. Test Scheme

The three asphalt mixtures, namely AC-25, AC-16 and AC-10, are formed according to the target grading used in the actual project as shown in Figure 3. In AC-13, AC-16 and AC-25, AC means asphalt concrete material. The numbers 13, 16 and 25 mean the nominal maximum size of aggregate. Nominal maximum size of aggregate refers to the ability of the aggregate to fit all through or to have a small amount that does not fit through (allowed to screen residual not more than 10%) the minimum standard sieve hole size, in mm. The design asphalt dosage is 4.52%, 4.16% and 3.83%, respectively, with designed porosity ranging between 3% and 6%. The shell 70# matrix asphalt is used as the formed asphalt, and limestone is used as the aggregate and mineral powder. Asphalt mixture specimens of 100 mm (diameter) × 84 mm (height) are produced though rotary compaction. There are 10 specimens with different voids for each grading, meaning a total of 30 rotating compaction specimens are involved as shown in Figure 4a.

A Percometer instrument is used to measure the dielectric constant of asphalt mixture specimen as shown in Figure 4b, which is composed of a host and a sensor probe. The test principle is to use two rings of metal inside and outside the sensor to form capacitance. By comparing the capacitance changes under the condition of taking the asphalt mixture test piece as the medium and the condition of taking the air as the medium, the relative dielectric constant of the asphalt mixture test piece is calculated according to Equation (4).
(4)$$\varepsilon =\frac{C^{\prime }}{C_{0}}$$

In this equation, $C^{\prime }$ is the capacitance measured with the material to be tested as the dielectric, and $C_{0}$ is the capacitance measured with the air as the dielectric. The relative dielectric constant test range of Percometer instrument is 1~32, the accuracy being ±0.1 and the detection frequency 40~50 mhz. The effective detection depth is 2~3 cm. Before the test, the top and bottom surface of the asphalt mixture test piece shall be cut 7 mm, respectively, to make both sides smooth and flat, so that the instrument sensor probe can closely fit the asphalt mixture test piece as shown in Figure 4c. The probe is hung in the air to test the relative dielectric constant of the air, calibrate the built-in parameters of the instrument and confirm that the instrument works smoothly, after which the probe is closely combined with the surface of the flat and smooth specimen to test and record the dielectric constant of the asphalt mixture specimen one by one. After the test, the surface dry method shall be used to measure and record the void ratio of the cut specimen as shown in Figure 4d.

### 3.2. Test Results and Analysis

The dielectric constant and porosity results of AC-13, AC-16 and AC-25 asphalt mixtures are plotted in a scatter diagram, while the relationship between the two variables is analyzed through linear regression as shown in Figure 5. The linear fitting formulas and correlation coefficients of AC-13, AC-16 and AC-25 are obtained in turn, as shown in Equations (5)–(7)
(5)$$v=-0.04545\varepsilon +0.33992\;R^{2}=0.81$$
(6)$$v=-0.03022\varepsilon +0.25037\;R^{2}=0.60$$
(7)$$v=-0.03593\varepsilon +0.29398\;R^{2}=0.49$$

In these equations, v is the void fraction, and ε is the relative dielectric constant.

The correlation coefficients between the relative permittivity and void fraction measured by AC-13, AC-16 and AC-25 are 0.81, 0.60 and 0.49, respectively, which has demonstrated a negative correlation. With the increase in the nominal maximum size of aggregate, the correlation coefficient shows a downward trend. The reason is that the asphalt mixture is an uneven mass composed of a variety of materials. In addition to the air component, the change of asphalt component, aggregate component and other factors will also cause the change of the dielectric constant of the mixture. The asphalt content and gradation variability of the mixture with a larger nominal maximum size is greater than that of the mixture with a smaller nominal maximum size, so the dielectric constant variability is also greater.

The main judgment basis for low compaction level segregation of asphalt mixture is that the void ratio of mixture exceeds the upper limit of the design void ratio threshold. The equation obtained by linear fitting is substituted into the critical value of low compaction level segregation of void fraction (*n* = 0.06), and the corresponding values ε_1_ of AC-13, AC-16 and AC-25 are calculated to be 6.20, 6.31 and 6.45, respectively. According to the void fraction test results of core samples, the proportion of low compaction level segregation of asphalt mixture of AC-13, AC-16 and AC-25 having a dielectric constant lower than ε_1_ is 100%, 100% and 66.7%, respectively. Therefore, it can be qualitatively considered that where the dielectric constant value ε is less than ε_1_, there is a large risk of low compaction level segregation, and this part of the point is labeled as a low compaction level segregation risk point (referred to as the risk point). The critical value of the dielectric constant should be measured separately by laboratory tests for asphalt mixtures of different levels and materials.

## 4. Calculation Method of Regional Indicators

The construction of new asphalt pavement has regional characteristics. For a small range of asphalt pavement area, the void ratio, asphalt content, gradation variation and other factors have certain relevance. Therefore, the more points whose dielectric constant test results are less than the threshold, the greater the probability of insufficient compaction in this region. However, the influence of the above regional dielectric constant test results on the regional low compaction level segregation degree is a qualitative expression. In order to quantitatively describe this influence, it is necessary to establish a relationship model between the regional dielectric constant test result characterization index and the regional low compaction level segregation degree characterization index. We consider the question of how to model relationships as follows:

1. A region is made up of points. We need to first determine which points are more likely to have insufficient compaction from the perspective of a single point. To solve the dielectric constant is to divide the risk points.

2. As discussed before, for a small area of asphalt pavement, the compaction level has a strong correlation, but the question is, how are we to divide the scope of each area? In step 1, we determined the risk points with insufficient compaction degree. To determine the risk areas with insufficient compaction degree, we can refer to the determination method of connected areas in digital images and take the connected risk points as an independent region analysis. We defined this region as the connected risk region; see Section 4.1 for details.

3. In addition, there are many “gaps” in the middle of each connected risk area, which are areas surrounded by connected risk points on four or three sides. In the digital image, they are called “lakes” and “bays”, as shown in Figure 6.

Although their permittivity detection values are smaller than the critical value, their compactness is strongly correlated with adjacent risk points. Moreover, in asphalt pavement construction technology adjustment and segregation repair, it is impossible to adjust and repair a single point. It is necessary to carry out construction technology adjustment and construction for a certain area together with the surrounding area. Therefore, with the help of the definition of the “convex hull” in digital images, the convex hull of the connected risk region is determined, and the connected risk region and the “gap” region surrounded by them are considered together as a whole; see Section 4.2 for details.

4. The question remains: How do we determine the quantitative relationship between the convex hull of the connected risk region and the degree of regional compaction insufficiency? A model needs to be established between the two indicators. The evaluation index of regional compactness can be cored at a certain spacing in the region to test the porosity of each core sample, which can be evaluated by the mean porosity of the core sample. The indexes that may be correlated between the convex hull of the connectivity risk region and the mean of the core void include the mean of the regional dielectric constant, the area of the region, the proportion of the risk points of the region, etc. A sensitivity analysis is needed to determine the correlation between them. See Section 5 for details.

### 4.1. Division of Connected Risk Areas

According to the dielectric constant threshold determined by the indoor test, the test results are divided into risk points and non-risk points. The distribution map of low compaction level segregation risk area based on dielectric constant is drawn by setting the risk points to white, while non-risk ones are black, as shown in Figure 7.

With the help of common analysis methods of digital images and by referring to the definition of connected regions and taking eight-adjacency as the connectivity criterion, the set of risk points connected to each other can be defined as connected risk regions, and the risk points can be divided into several connected risk regions, as shown in Figure 8. The arrow refers to the connected risk region with the largest area product.

### 4.2. Determination of Convex Hull of Connected Risk Area

The connected risk area is usually an irregular complex polygon. In the actual construction process, the engineering treatment measures and construction process adjustment need to be continuously treated or adjusted in a large area so it is impossible to take precise and effective measures for this complex polygon. Therefore, one feasible way is to include the internal voids (lakes) and open voids (bays) connecting the risk area to form a convex area. In the digital image processing method, it is necessary to determine the convex hull of the connected risk region.

H is a convex region, which means that for region H, if and only if for any two points $x_{1},x_{2}\in H$, the entire straight line segment $x_{1},x_{2}$ defined by its endpoints $x_{1}x_{2}$ is located inside the region H. The convex hull of region R refers to the minimum convex region H that satisfies the condition where R includes H. The convex hull of a region is the minimum convex region satisfying the condition. As shown in Figure 8, the part enclosed by the red box is the convex hull of the connected risk area. It is mainly determined by detecting the edge points of the area, connecting the edge points in turn to form an enclosed area. The detection steps are shown in Figure 9 and are described in detail as follows:Locate the point $P_{0}$ with the smallest vertical coordinate, which is one of the convex hull edge points;From $P_{0}$, determine the convex hull edge points one by one in a counterclockwise direction;$P_{1}$ is determined by forming a vector with $P_{0}$ one by one except for the points that have been determined as convex hull edge points and calculating the included angle β between the vector and the horizontal line. The point with the smallest angle β is $P_{1}$;The determination method of $P_{i}(i>1)$ is as follows: the points other than those determined as convex hull edge points and $P_{i-1}$ form a vector one by one. After the included angle α between the vector and the extension line of vector $P_{i-2}P_{i-1}$ is calculated, the point with the smallest angle a is $P_{i}(i>1)$;Repeat step 4 until $P_{i}=P_{1}$.

### 4.3. Calculation of Regional Indicators

The characterization indicators of regional dielectric constant test results mainly include: connected risk area, convex hull area, convex hull risk point ratio and regional dielectric constant mean value. The general calculation method of area is based on the number of pixels. The calculation method is shown in Equation (8):(8)$$A=Nd_{x}d_{y}$$

In this equation, A is the area, N is the number of sampling points in the area, $d_{x}$ is the horizontal spacing of sampling points, that is, the sampling spacing of dielectric constant results and $d_{y}$ is the spacing in the width direction of sampling points.

The area ratio between the connected risk area and the regional convex hull is the convex hull risk point ratio, which is used to evaluate the proportion of risk points in the convex hull area and is calculated according to Equation (9).
(9)$$\alpha =\frac{A}{A_{k}}$$

In this equation, α is the ratio of convex hull risk points, A is the area of connected risk area and $A_{k}$ is the area of regional convex hull.

The mean value of regional dielectric constant is the mean value of dielectric constant test results at all risk points in the region.

## 5. Physical Engineering Test

### 5.1. Test Scheme

The physical engineering test was carried out on a 400 m AC-25 pavement of an expressway asphalt lower layer. The raw materials, grading curve and designed asphalt content consistent with the indoor test were used in the test section, and the designed value of void ratio was 3%~6%. The test section adopts the combined rolling scheme of static pressing once with 13T double steel wheel roller, vibrating pressing twice with 13T double steel wheel roller, static pressing three times with 30T rubber wheel roller and static pressing once with 13T rubber wheel roller. After the construction is completed, low and high compaction level aggregate accumulation areas can be seen on the surface, as shown in Figure 10b. After 24 h of construction, when the pavement temperature drops to a stable value, the three-dimensional ground-penetrating radar shall be used to collect the dielectric constant results of the test section immediately to avoid dust, rain and other debris from entering the asphalt pavement gap and interfering with the test results, as shown in Figure 10c. The test section is divided into two 200 m units for analysis, numbered 1 and 2. Unit 1 calculates the relevant indicators of its low compaction level segregation risk connectivity area, and studies the relationship between the regional indicators and the degree of low compaction level segregation. Unit 2, as the verification unit, verifies the relevant analysis conclusions of unit 1.

### 5.2. Test Results

According to the determination threshold of the AC-25 asphalt mixture low compaction level segregation risk points (6.45) determined in the indoor test, the distribution of segregation risk areas in unit 1 is drawn, as shown in Figure 11. There are 102 connected segregation risk areas in unit 1, with an area of 0.05~4.30 m^2^. Due to the excessive number of areas, a total of nine areas of 0.30~4.30 m^2^ were selected as the research objects at an interval of about 0.5 m^2^. Core samples were drilled at an interval of 1 m within each connected risk area. The surface dry method was used to measure the gross volume density of core samples and calculate their void-age, and the average void-age of regional core samples represents the degree of segregation in the area. The four regional indicators of connected risk area, regional convex hull area, convex hull risk point ratio and regional mean dielectric constant are listed in Table 1. Meanwhile, drill core samples at five non-risk points were randomly selected within the detection range to measure the void ratio of the core samples. The test results are summarized in Table 2.

In this form, $\bar{\varepsilon }$ is the mean value of regional dielectric constant, ν is the void ratio of the core sample, $\bar{\nu }$ is the void ratio of core sample, α is the convex hull risk point ratio (%), $A_{k}$ is the area of the convex hull (m^2^) and A is the connected risk area (m^2^).

The average void ratio of core samples drilled in the connected risk area is 6.4% while that of non-risk points is 4.2%. All of the void ratios of the core samples drilled in the non-risk areas do not exceed the upper limit of the designed void ratio threshold, whereas seven out of the nine connected risk areas exceed the upper limit of the design void ratio threshold. On the premise that the core void distribution obeys the normal distribution, the significance level (α=0.005) is taken to test the core void ratio in the hypothetically non-risk point area (n<0.06). After calculation and inspection, it shows that the risk of segregation at non-risk points is low ($\frac{\bar{z}-\mu_{0}}{S^{*}/\sqrt{n}}=\frac{0.424-0.6}{0.008295/\sqrt{4}}=-42.44<-4.60$). Similarly, the porosity of core samples in the connected risk area (n>0.06) is tested. The significance level α=0.005 is taken for calculation, and the results are greater than 4.6 ($\frac{\bar{z}-\mu_{0}}{S^{*}/\sqrt{n}}=\frac{0.638-0.6}{0.00618/\sqrt{8}}=17.39>4.60$). The test shows that the connected risk area has a high likelihood of segregation. Therefore, it can be considered that the risk of segregation in the connected risk area is much higher than that in the non-risk point area, and the connected risk area is mainly concerned in the study of segregation area detection.

Among the regional indicators, there is an algebraic relationship between the area of the connected risk region, the area of the convex hull and the ratio of the convex hull risk points. Therefore, the index of the area of the convex hull is discarded, and the relationship between the area of the connected risk region, the ratio of the convex hull risk points, the average value of the regional dielectric constant and the average value of the regional core void ratio are analyzed by response surface analysis. The relationship between the regional indicators of connected risk areas and the degree of regional segregation is studied.

### 5.3. Regression Model Fitting and Variance Analysis

Design-Expert 12 is used to perform regression fitting on the data in Table 1, and the regression model equation is shown in Equation (10),
(10)$$v_{p}=-152.748+0.457\times S+2.341\times \alpha +26.442\times \bar{\varepsilon }-0.018\times S\times \alpha +0.191\times S\times \bar{\varepsilon }-0.391\times \alpha \times \bar{\varepsilon }$$

In this equation, v is the predicted mean value of regional core void ratio, $\bar{\varepsilon }$ is the mean value of regional dielectric constant, S is the area of connected risk area and α is the ratio of convex hull risk points.

Table 2 shows the variance analysis of the mean void content of regional core samples. The F value of regression model is 302.38; the p value is 0.0033, which demonstrates high significance; and the $R_{Adj}^{2}$ value is 0.9820, indicating that the correlation between the observed value and the predicted value of the equation is good, the error is small and the model fitting effect is good. According to the F value, the order of the influence of each factor on the mean value of regional core void ratio is as follows: mean value of regional dielectric constant > ratio of convex hull risk points > area of connected risk area.

According to the results of regression analysis, corresponding response surface and contour maps are drawn, as shown in Figure 12, Figure 13 and Figure 14.

The interaction between various factors can be seen more intuitively in the response surface diagram and contour map. The steeper the surface, the greater the influence of factors on the results. The ellipse of contour lines indicates that the interaction between factors is obvious, and vice versa. It can be seen from Figure 12 that the slope of the response surface diagram of the ratio of the area of the connected risk area to the risk point of the convex hull to the void ratio of the core sample changes slightly, and the curvature of the contour line is general, indicating that the factors have an impact on the results, and the interaction between the factors is not significant.

It is observable from Figure 13 that the slope change of the response surface diagram of the area of the connection risk area and the mean value of the regional dielectric constant to the core void ratio is marginal, and the curvature of the contour line is small, revealing that the factors have an impact on the results, and the interaction among the factors is not significant. What can be found from Figure 14 is that the slope of the response surface diagram of the mean dielectric constant and the ratio of convex hull risk points in the connected area to the core sample void ratio shows substantial change, and the contour curve is huge, meaning that the factors have an impact on the results and the interaction among the factors is significant.

### 5.4. Model Validation

The distribution of segregation risk areas in unit 2 is shown in Figure 15. There are 118 connected segregation risk areas in unit 2. According to the regression model of Equation (10), the predicted void ratio is 5.5%~7.5%. At an interval of about 0.2%, 11 areas were selected as the verification area. Core samples were drilled in the verification area at an interval of 1 m. The surface dry method was used to measure the gross volume density of core samples, calculate the void ratio and the regional indicators of the verification area, and the test results of core samples are listed in Table 3.

The verification results in Table 2 show that the error range between the predicted voids and the measured voids is −0.4%~+0.4%, with the mean absolute value of the error being 0.25% and the relative error 3.77%, compared with the mean measured voids of 6.63%, indicating that the model can accurately predict the regional segregation degree.

## 6. Conclusions

In this paper, the response surface analysis method is used to regression fit the regional indicators and regional low compaction level segregation degree of connected risk areas. The main conclusions are as follows:(1)Through indoor tests, the correlation coefficients between measured relative dielectric constants and voids of AC-13, AC-16 and AC-25 asphalt mixtures are 0.81, 0.60 and 0.49 by linear fitting. It shows that there is a negative correlation between them, and the correlation coefficient shows a downward trend with the increase in the maximum nominal particle size, which is mainly related to the influence of other components except air on the dielectric constant of asphalt mixture. The equation obtained by linear fitting is substituted into the upper limit of the design porosity threshold, and the corresponding dielectric constant threshold $\varepsilon_{1}$ is calculated. It can be qualitatively considered that the point where the dielectric constant ε is less than $\varepsilon_{1}$ has a greater risk of low compaction level segregation.(2)According to the dielectric constant threshold determined by the indoor test, the test results are divided into risk points and non-risk points. The set of inter-connected risk points is defined as a connected risk area. The convex hull of the connected risk area is detected, and the voids of nine connected risk areas and five non-risk points are tested by coring at the test section of entity project unit 1. On the premise that the void ratio distribution of core sample obeys the normal distribution, the significance test of α=0.005 shows that the risk of low compaction level segregation in the connected risk area is much higher than that in the non-risk point area.(3)Three regional index calculation method is proposed, involving the area of the connected risk region, ratio of risk points of the convex hull and mean value of the regional dielectric constant. The three factors and the mean value of regional core voids are put into regression model using the response surface method. The F value of the regression model is 302.38, and the p value is 0.0033, showing that the result is highly significant. The $R_{Adj}^{2}$ is 0.9820, indicating that the correlation between the observed values and the predicted values of the equation is positive with small error and ideal model-fitting effect. According to the F value, the order of the influence of each factor on the mean value of regional core void ratio is as follows: mean value of regional dielectric constant > ratio of convex hull risk points > area of connected risk area.(4)The model obtained by the regression fitting of the response surface method selects 11 areas in the unit 2 as the verification areas. The predicted porosity is calculated according to the regional indicators. The error range between the predicted porosity and the measured porosity is −0.4%~+0.4%, and the average absolute value of the error is 0.25%. The results are compared with the measured average porosity of 6.63%, and the relative error is 3.77%, indicating that the model can more accurately predict the degree of regional low compaction level segregation.

Research manuscripts reporting large datasets that are deposited in a publicly available database should specify where the data have been deposited and provide the relevant accession numbers. If the accession numbers have not yet been obtained at the time of submission, please state that they will be provided during review. They must be provided prior to publication.

Interventionary studies involving animals or humans, and other studies that require ethical approval, must list the authority that provided approval and the corresponding ethical approval code.

## Figures and Tables

**Figure 1 sensors-22-07980-f001:**
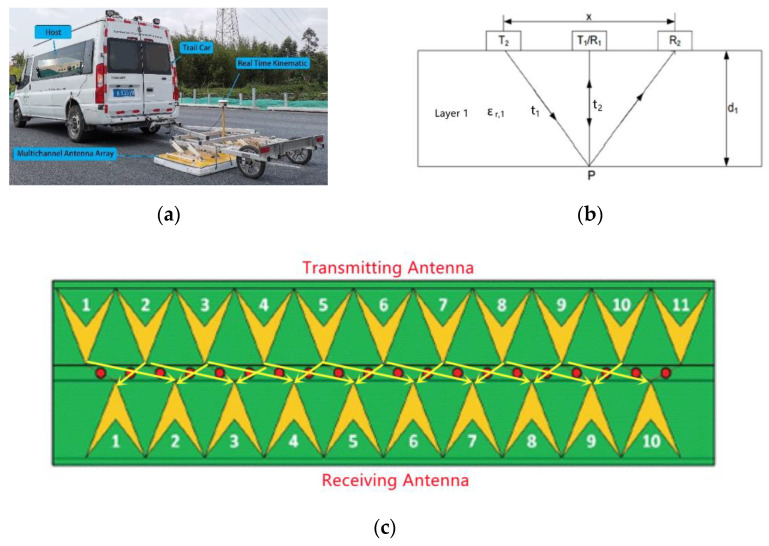
Onboard three-dimensional ground-penetrating radar system and detection principle. (**a**) Onboard three-dimensional ground-penetrating radar system. (**b**) Propagation model of electromagnetic wave for dielectric constant detection. (**c**) Antenna combination configuration diagram.

**Figure 2 sensors-22-07980-f002:**
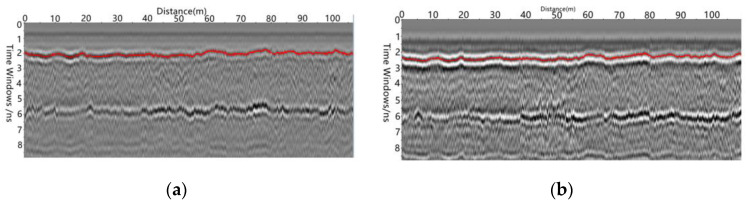
Scanning radar image by common mid-point method. (**a**) Combined asphalt base interface reflection signal of adjacent antennas. (**b**) Reflection signal of asphalt base interface of antenna combination at a distance of X meters.

**Figure 3 sensors-22-07980-f003:**
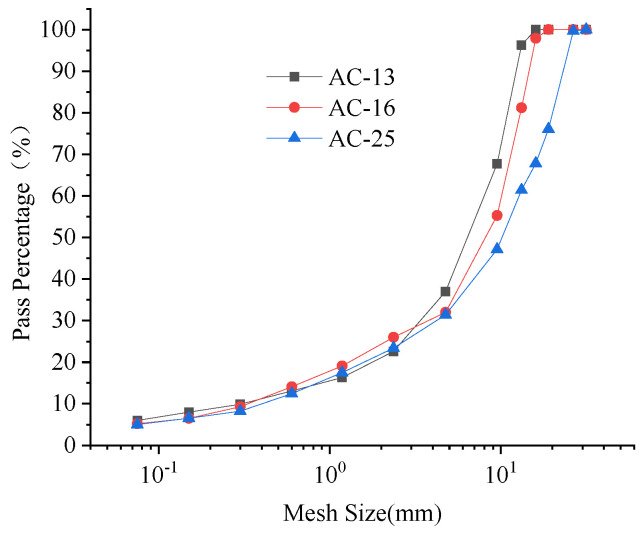
Grading curve of formed specimen.

**Figure 4 sensors-22-07980-f004:**
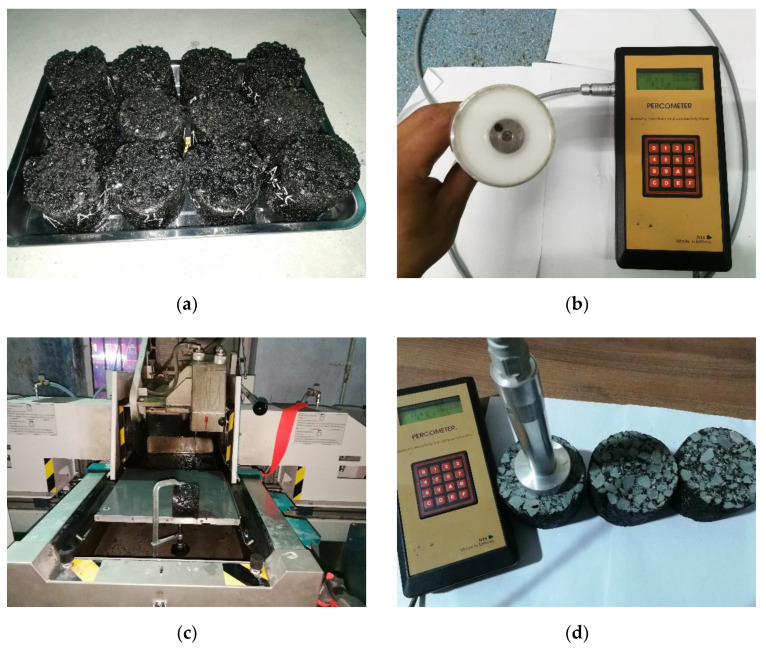
Measurement of dielectric constant of test piece. (**a**) Rotating compacted specimen. (**b**) Percometer instrument. (**c**) Specimen cutting. (**d**) Dielectric constant measurement of test piece.

**Figure 5 sensors-22-07980-f005:**
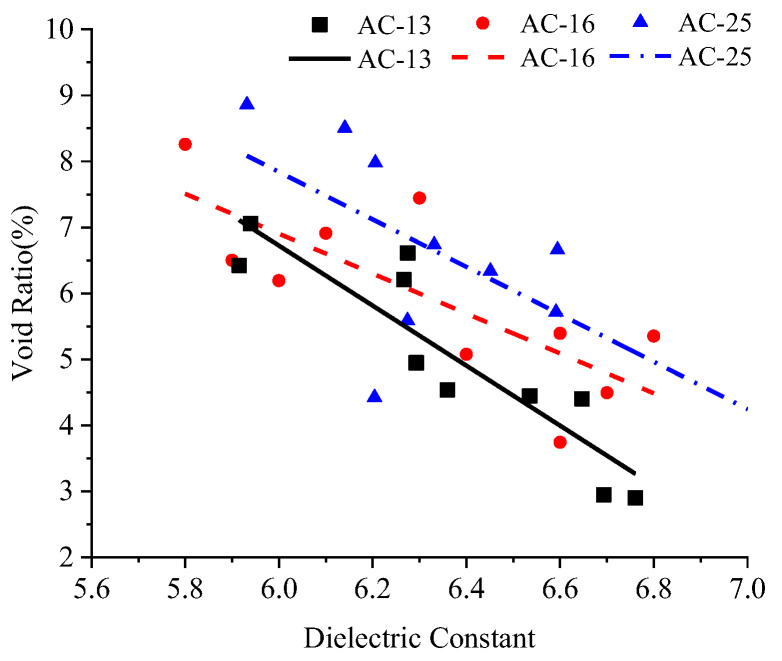
Distribution diagram of void fraction and dielectric constant of test piece.

**Figure 6 sensors-22-07980-f006:**
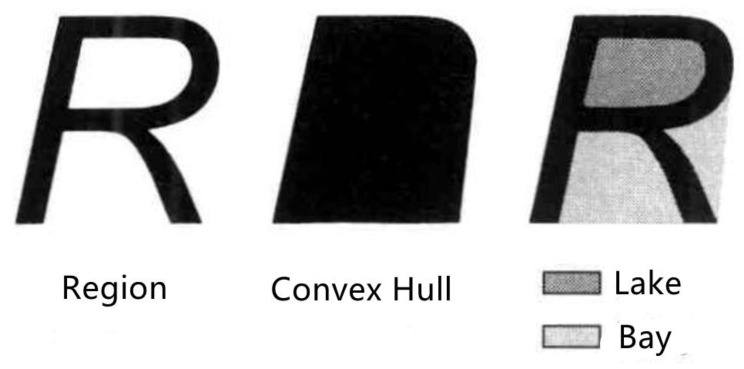
Object “R”, its convex hull and its associated lakes and bays.

**Figure 7 sensors-22-07980-f007:**
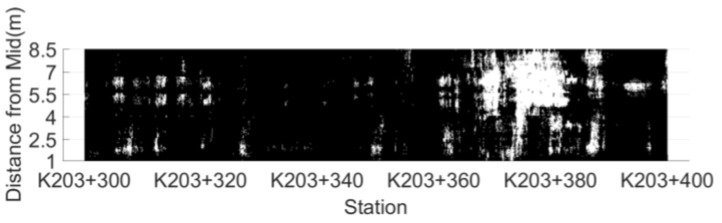
The distribution map of low compaction level segregation risk area.

**Figure 8 sensors-22-07980-f008:**
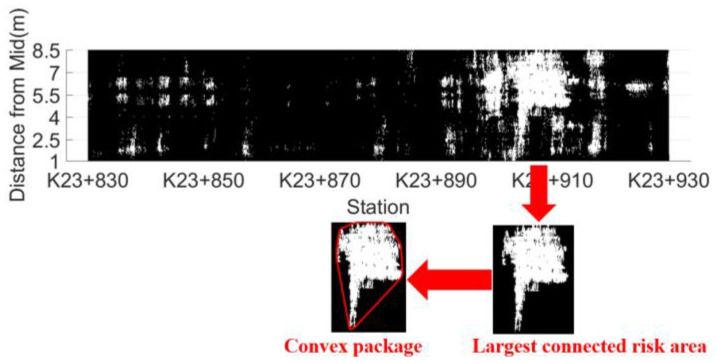
Regional distribution of segregation risk areas.

**Figure 9 sensors-22-07980-f009:**
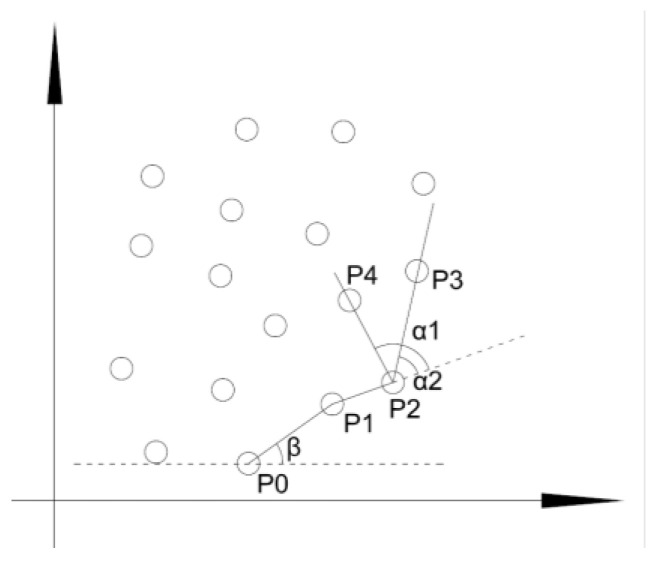
Convex hull determination method of connected risk region.

**Figure 10 sensors-22-07980-f010:**
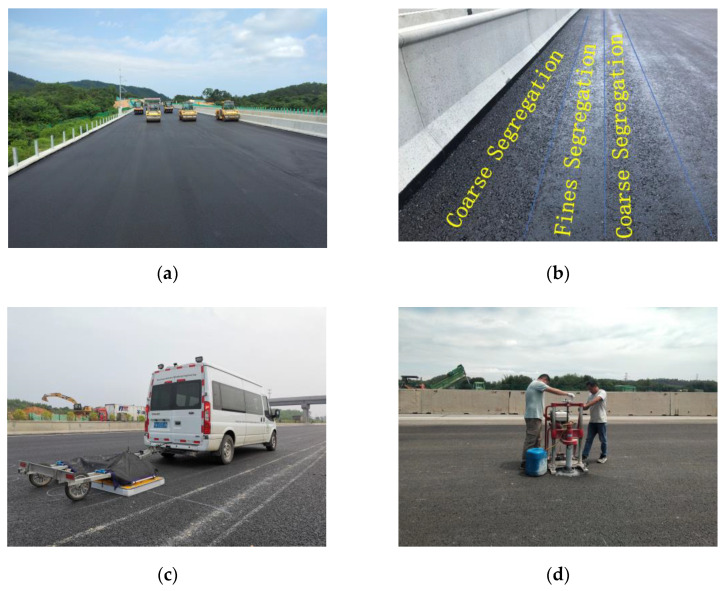
Physical engineering test. (**a**) Construction of the lower layer of asphalt. (**b**) Strip agglomeration area of low and high compaction level aggregates on the surface. (**c**) 3D GPR scanning. (**d**) Void ratio measurement in low compaction level segregation risk area.

**Figure 11 sensors-22-07980-f011:**
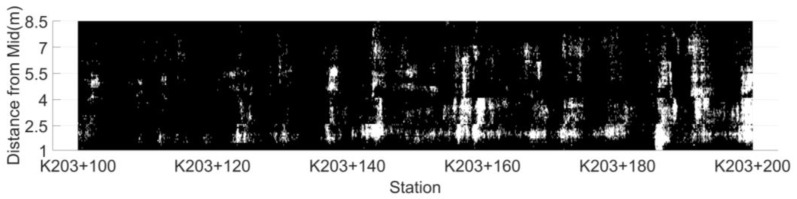
Distribution of connected risk areas in unit 1.

**Figure 12 sensors-22-07980-f012:**
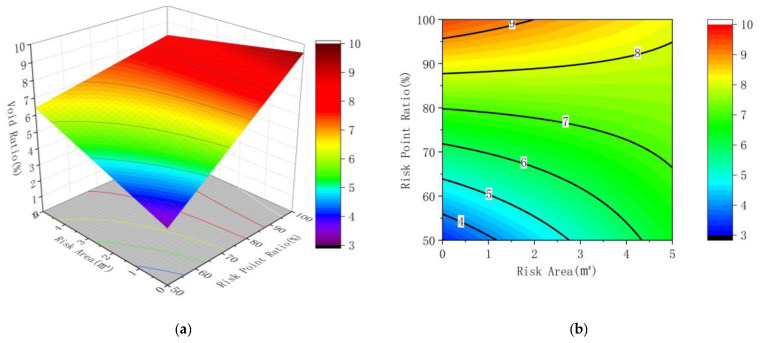
Response surface diagram and contour diagram of the risk point ratio and the risk area to the void ratio; (**a**) the response surface diagram (**b**) the contour diagram.

**Figure 13 sensors-22-07980-f013:**
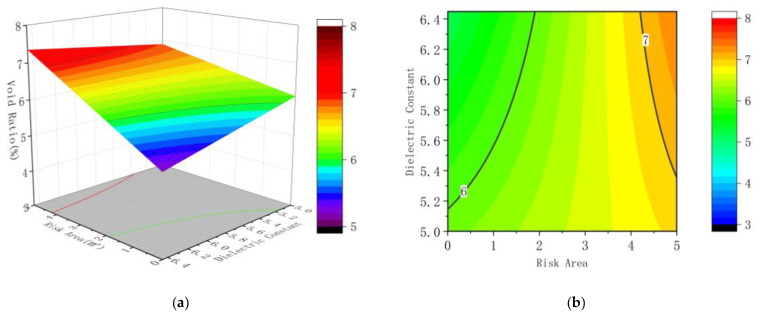
Response surface diagram and contour diagram of the dielectric constant and the risk area to the void ratio; (**a**) the response surface diagram (**b**) the contour diagram.

**Figure 14 sensors-22-07980-f014:**
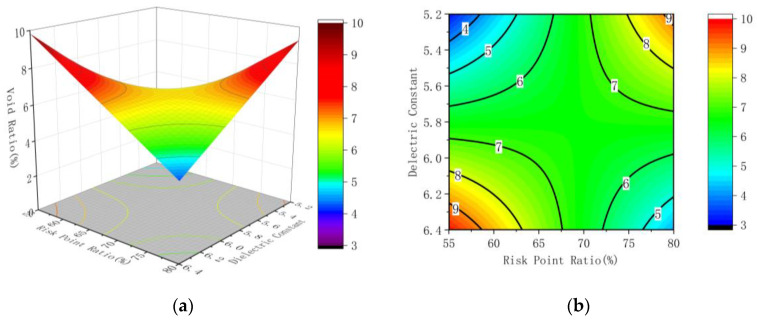
Response surface diagram and contour diagram of the risk point ratio and the delectric constant to the void ratio; (**a**) the response surface diagram (**b**) the contour diagram.

**Figure 15 sensors-22-07980-f015:**
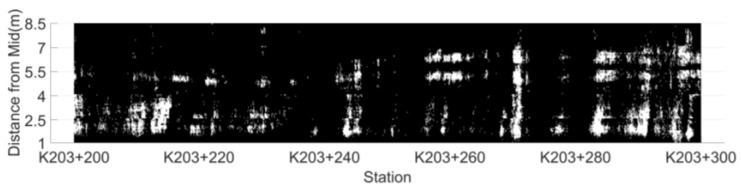
Distribution map of connected risk areas in unit 2.

**Table 1 sensors-22-07980-t001:** Regional indicators of connected risk areas.

No.	A	$A_{k}$	α	$\bar{\varepsilon }$	ν					$\bar{\nu }$
1	4.30	5.76	74.7%	5.50	6.6%	7.3%	7.9%	8.2%		7.5%
2	3.72	6.83	54.5%	5.69	6.2%	6.4%	5.9%	5.4%		6.0%
3	3.22	4.86	66.4%	5.47	6.3%	5.9%	6.8%			6.3%
4	2.78	3.88	71.7%	5.26	7.3%	6.7%	6.9%			7.0%
5	2.04	3.92	52.1%	5.82	5.9%	5.6%				5.8%
6	1.72	2.46	69.9%	5.79	5.7%	6.3%				6.0%
7	1.10	1.40	78.4%	5.72	6.8%					6.8%
8	0.71	0.93	75.9%	5.75	6.4%					6.4%
9	0.30	0.38	79.3%	5.96	5.6%					5.6%
10	non-risk point		7.12	3.8%	3.3%	4.9%	5.3%	3.9%	4.2%	7.12

**Table 2 sensors-22-07980-t002:** The variance analysis of the mean void content of regional core samples.

Source	Sum of Squares	df	Mean Square	F-Value	*p*-Value	
Model	3.90	6	0.649	302.38	0.0033	significant
A-Risk Area	0.025	1	0.025	11.53	0.077	
B-Risk Point Ratio	0.777	1	0.777	361.81	0.0028	
C-Average Dielectric Constant	1.00	1	1.00	465.88	0.0021	
AB	0.148	1	0.148	69.08	0.0142	
AC	0.008	1	0.0082	3.83	0.190	
BC	0.482	1	0.485	225.73	0.0044	
Residual	0.0043	2	0.0021			
$R^{2}$	0.9955					
$R_{Adj}^{2}$	0.9820					

**Table 3 sensors-22-07980-t003:** The regional indicators of the verification area and the test results of core samples.

No.	A	α	$\bar{\varepsilon }$	$\nu_{p}$	ν	e
1	3.89	74.4%	5.49	7.5%	7.3%	0.2%
2	2.23	81.0%	5.69	7.3%	7.7%	−0.4%
3	4.85	58.3%	5.74	7.1%	6.9%	0.2%
4	3.51	61.9%	5.84	6.9%	6.7%	0.2%
5	2.40	73.2%	5.69	6.8%	7.0%	−0.2%
6	0.68	91.6%	5.89	6.5%	6.3%	0.2%
7	0.11	84.2%	5.87	6.3%	6.5%	−0.1%
8	1.36	74.6%	5.90	6.1%	5.8%	0.3%
9	0.70	70.0%	5.87	5.9%	6.1%	−0.3%
10	1.65	60.6%	5.74	5.7%	6.0%	−0.4%
11	0.42	78.8%	6.02	5.5%	5.1%	0.4%

## Data Availability

Not applicable.

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
