# Peer review of "Low Compaction Level Detection of Newly Constructed Asphalt Pavement Based on Regional Index"

_sensors, 2022, doi:10.3390/s22207980_

Round 1
Reviewer 1 Report
Dear authors,
Your article is very interesting, timely and relevant to the industry. I commend you for picking such a complex but essential topic and for your fine work. However, in my view, this present paper in its current version is not presented well and doesn't do good service to the work you have carried out in the study. In my view, it needs major technical and editorial revision. Please revise the terminologies you employed to ensure they are understood internationally as you intended to. Refrain from frequently using terminologies and computation approaches that you have not properly introduced and with described the goal.
Most importantly, the paper does not convincingly argue that what you are measuring is coarse segregation. In my opinion, what you are observing and measuring could very well be the change of dielectrics due to different compaction levels, not coarse segregation. If you are measuring the level of compaction and state so: the topic will remain novel and relevant. For these reasons, I think it will benefit both the industry, the academia and yourselves if you put more work into this paper. I have listed more specific comments from the paper below
Comments
Page 2 - lines 46 to 48: It is unclear how coring affects the integrity of the subgrade and pavement structure as long the holes are filled immediately. I understand the damage to the asphalt layer, but the damage to the subgrade and pavement structure is not clear from this paragraph. It would help if you elaborated on this further.
Page 2 – line 63: Reconsider the sentence "Most researchers". Have you considered all the researchers and determined that most of them fall in the category? I would consider removing most. The number of researchers using GPR is limited.
Page 2 – lines 77: what is independent of what? The sentence has only one subject: the final prediction results proposed by rapid and NDT technology". It is unclear what you mean by "are still independent of each other."
Page 2 – lines 77 to 90. The entire paragraph needs to be revised and simplified. I am not sure I understand the meaning of the terms "regionality", "regional indicator", "regional perspective" and "regional test results" in this context.
Page 3, lines 98 to 103: In this paragraph, you used many new terms (i.e., convex hull, regional indicator, regional area, surface analysis) without adequately introducing them. For this reason, this paragraph confuses readers. Don't assume all readers are experts on the work you are presenting. Consider re-writing this paragraph
Page 3 line 109: With a 1.5 m antenna array, you probably want to cover a road's full-section. May be one or half lane. Your example in Figure 1, shows you not covering the entire section. Correct to full-lane or partial-lane coverage.
Page 3, lines 110 to 111: You didn't indicate the antenna model used in this study, so I will make some assumptions. The 0.071 m spacing is between the receivers. From the picture in Figure 1, it seems like you are using a Geoscope Radar antenna from 3D-Radar. If these assumptions are correct, you will have 11 pairs of receivers that you can configure in many different ways for the data collection. It is important for the flow of the paper that you indicate how you configured your antenna for the common midpoint collection. what are are the apscing betwen the transmitter and reciever? How may pairs you considered?. These are important piece of information that should have been included
Page 3 line 119: "There are three main methods for the dielectric constant of asphalt"? Did you mean "There are three main methods for determining the dielectric constant of asphalt pavement"?
Page 4 line 146: I can't find Figure 1C
Page 3 Figure 2. I am assuming that the difference you wanted to highlight between figures a and b is the time depth values in the Y axes. Unfortunately, the values are too small to read. You should explain in the body of the paper or choose better figures.
Page 4 line 148. I am assuming by the term "detection" you mean survey or scan. Consider using these terms instead.
What do you mean by "x can be measured before detection"? x should be known: it is the distance between the receivers that you selected. You don't need to measure it before the survey.
Page 4 line 154: Figure 3 doesn't have any information on gradation as you say
Page 6 line 195: You didn't provide the maximum nominal particle size of the mixtures. You mention them in the paragraph without reporting them. Revise
Page 6 Indoor test/test results. If you have the chance, consider determining the laboratory dielectrics using a more reliable and accurate GPR system than a percometer. The GSSI pavescan system have a very robust system for measuring dielctric of asphalt from laboratory specimens through the surface reflection or the time-of-flight method. Both of them produce excellent relationships between void content and dielectric. This may strengthen the void vs diectrict relationships in your paper and the determination of the critical dielectric values as you explained in paragraph 203 to 211 of page 6. You can find more information in this paper "Ground Penetrating Radar Sensitivity to Marginal Changes in Asphalt Mixture Composition"
Here is something that is not clear in the article. There is no doubt that there is a strong relationship between the void content and the dielectric constant of asphalt mixtures. It has been demonstrated in many research studies. The same mixture compacted at different levels of energy will have different void and dielectrics. You hint at this in Page 7 lines 216 to 217. Both the laboratory approaches and the filed surveys you discuss do not convincingly argue that what they you are measuring is coarse segregation. In my opinion, what you are observing and measuring could very well be the change of dielectrics due to different compaction levels, not coarse segregation.
Page 7 Section 4: The entire sections needs to be revised and re-written. I found it very difficult to understand the objective and and the process applied in this calculation. Perhaps you should start by introducing the techniques and objective soughts.
Page 9 lines 295 to 298: Dielectric constant is not a function of temperature. Why wait until the temperature is stable? What not test it immediately after construction so that you can fix damaged area's immediately
Author Response
Dear Editors and Reviewers:
Thank you for your letter and for the reviewers’ comments concerning our manuscript"entitled “Low Compaction Level Detection of Newly Constructed Asphalt Pavement Based on Regional Index"(ID:1891355). Those comments are all valuable and very helpful for revising and improving our paper, as well as the important guiding significance to our researches. We have studied comments carefully and have made correction which we hope meet with approval. Revised portion are marked in red in the paper. The main corrections in the paper and the responds to the reviewer's comments are as flowing:
1. Page 2 - lines 46 to 48: It is unclear how coring affects the integrity of the subgrade and pavement structure as long the holes are filled immediately. I understand the damage to the asphalt layer, but the damage to the subgrade and pavement structure is not clear from this paragraph. It would help if you elaborated on this further.
Response: We are very sorry that we have neglected to elaborate the mechanism of coring damage to the pavement structure. Relevant content has been further supplemented. As shown in the revised manuscript Page 2 - lines 48 to 53.
2. Page 2 – line 63: Reconsider the sentence "Most researchers". Have you considered all the researchers and determined that most of them fall in the category? I would consider removing most. The number of researchers using GPR is limited.
Response: Considering the Reviewer’s suggestion, "Most" has been removed. As shown in the revised manuscript Page 2 - lines 69.
3.Page 2 – lines 77: what is independent of what? The sentence has only one subject: the final prediction results proposed by rapid and NDT technology". It is unclear what you mean by "are still independent of each other."
Response: The sentence was supposed to mean ”At present, the final prediction results proposed by the rapid and nondestructive testing technology for low compaction level segregation of asphalt pavement are still evaluated separately according to the single point test results, which are completely independent of each other without considering the correlation between adjacent points. However, during the construction of asphalt pavement, the distribution of construction quality often presents a certain regionality, that the aggregate gradation, paving temperature, compaction temperature, compaction power, etc. between two close points are relatively similar (normally within 1m).” As shown in the revised manuscript Page 2 - lines 83-90.
4. Page 2 – lines 77 to 90. The entire paragraph needs to be revised and simplified. I am not sure I understand the meaning of the terms "regionality", "regional indicator", "regional perspective" and "regional test results" in this context.
Response: We are very sorry for our incorrect writing. "regionality" means during the construction of asphalt pavement, the distribution of construction quality often presents a certain regionality, that the aggregate gradation, paving temperature, compaction temperature, compaction power, etc. between two close points are relatively similar (normally within 1m). In order to avoid the reader's confusion, the paragraphs have been simplified and only a brief introduction of the research means and objectives has been retained. As shown in the revised manuscript Page 3 - lines 100-103.
5. Page 3, lines 98 to 103: In this paragraph, you used many new terms (i.e., convex hull, regional indicator, regional area, surface analysis) without adequately introducing them. For this reason, this paragraph confuses readers. Don't assume all readers are experts on the work you are presenting. Consider re-writing this paragraph
Response: We have re-written this part according to the Reviewer's suggestion. In order to avoid the reader's confusion, the paragraphs have been simplified and only a brief introduction of the research means and objectives has been retained. As shown in the revised manuscript Page 3 - lines 100-103.
6.Page 3 line 109: With a 1.5 m antenna array, you probably want to cover a road's full-section. May be one or half lane. Your example in Figure 1, shows you not covering the entire section. Correct to full-lane or partial-lane coverage.
Response: In the article, the phrase "By setting multiple detection channels, the full cross-section coverage scanning of any road can be realized." is mentioned.As shown in the revised manuscript Page 3 - lines 109-110.
Figure 1 is a schematic of the field test, which is used to introduce the composition of the 3D GPR equipment.
7.Page 3, lines 110 to 111: You didn't indicate the antenna model used in this study, so I will make some assumptions. The 0.071 m spacing is between the receivers. From the picture in Figure 1, it seems like you are using a Geoscope Radar antenna from 3D-Radar. If these assumptions are correct, you will have 11 pairs of receivers that you can configure in many different ways for the data collection. It is important for the flow of the paper that you indicate how you configured your antenna for the common midpoint collection. what are are the apscing betwen the transmitter and reciever? How may pairs you considered?. These are important piece of information that should have been included
Response: It is really true as Reviewer suggested that we use a Geoscope Radar antenna from 3D-Radar. The collection parameter Settings used in the detection are introduced in the revised manuscript Page 3 - lines 110-112. Meanwhile, the antenna combination configuration of the common midpoint method is explained in detail in the revised manuscript Page 3 - FIG. 1 (c) and Page 4 - lines 134-135.
8. Page 3 line 119: "There are three main methods for the dielectric constant of asphalt"? Did you mean "There are three main methods for determining the dielectric constant of asphalt pavement"?
Response: It is really true as Reviewer suggested that “There are three main methods for the dielectric constant of asphalt pavement by GPR”.As shown in the revised manuscript Page 3 - lines 120-121.
9. Page 4 line 146: I can't find Figure 1c
Response: “Figure 1c” has been modified to “Figure 2b”.As shown in the revised manuscript Page 4 - lines 149.
10. Page 3 Figure 2. I am assuming that the difference you wanted to highlight between figures a and b is the time depth values in the Y axes. Unfortunately, the values are too small to read. You should explain in the body of the paper or choose better figures.
Response: It is really true as Reviewer suggested that the time depth values in the Y axes are too small to read. the Y-axis values of Figure 2 have been changed to a larger font. As shown in the revised manuscript Page 3 - Figure 2.
11. Page 4 line 148. I am assuming by the term "detection" you mean survey or scan. Consider using these terms instead.
Response: As Reviewer suggested that the term "detection" was corrected as “scan”.As shown in the revised manuscript Page 4 - lines 154.
12. What do you mean by "x can be measured before detection"? x should be known: it is the distance between the receivers that you selected. You don't need to measure it before the survey.
Response: As Reviewer suggested that x is known: it is the distance between the receivers that you selected. As shown in the revised manuscript Page 4 - lines 152.
13. Page 4 line 154: Figure 3 doesn't have any information on gradation as you say.
Response: We are very sorry for our incorrect writing. "Figure 3" was corrected as “Figure 4”.As shown in the revised manuscript Page 4 - lines 159.
14. Page 6 line 195: You didn't provide the maximum nominal particle size of the mixtures. You mention them in the paragraph without reporting them. Revise.
Response: We are very sorry that we have neglected to elaborate the maximum nominal particle size of the mixtures. Nominal maximum size of aggregate refers to the aggregate can all through or a small amount of not through (allowed to screen residual not more than 10%) the minimum standard sieve hole size, in mm. Related explanations have been added to the revised manuscript Page 4 - lines 159-163.
15. Page 6 Indoor test/test results. If you have the chance, consider determining the laboratory dielectrics using a more reliable and accurate GPR system than a percometer. The GSSI pavescan system have a very robust system for measuring dielctric of asphalt from laboratory specimens through the surface reflection or the time-of-flight method. Both of them produce excellent relationships between void content and dielectric. This may strengthen the void vs diectrict relationships in your paper and the determination of the critical dielectric values as you explained in paragraph 203 to 211 of page 6. You can find more information in this paper "Ground Penetrating Radar Sensitivity to Marginal Changes in Asphalt Mixture Composition".
Response: As Reviewer suggested that the GSSI pavescan system have a very robust system for measuring dielctric of asphalt from laboratory specimens through the surface reflection or the time-of-flight method, but we don't have a system in the lab, and we can't borrow it. As the research results in the paper “Complex permittivity measurement using capacitance method from 300 kHz to 50 MHz”, Measurement obtained using the parallel-plate capacitance was compared with the free space method to validate its accuracy. The percent difference is less than 5%. The Percometer instrument based on the principle of parallel plate capacitance method is a reliable method to measure the dielectric constant of materials.
16. Here is something that is not clear in the article. There is no doubt that there is a strong relationship between the void content and the dielectric constant of asphalt mixtures. It has been demonstrated in many research studies. The same mixture compacted at different levels of energy will have different void and dielectrics. You hint at this in Page 7 lines 216 to 217. Both the laboratory approaches and the filed surveys you discuss do not convincingly argue that what they you are measuring is coarse segregation. In my opinion, what you are observing and measuring could very well be the change of dielectrics due to different compaction levels, not coarse segregation.
Response: We are very sorry for our incorrect writing. The use of "coarse segregation" is ambiguous and it was corrected as “Low Compaction Level” in the revised manuscript.
17.Page 7 Section 4: The entire sections needs to be revised and re-written. I found it very difficult to understand the objective and and the process applied in this calculation. Perhaps you should start by introducing the techniques and objective soughts.
Response: It is really true as Reviewer suggested that we should start by introducing the techniques and objective soughts. We introduce in detail the questions, starting points and ideas of the research of Section 4 and 5 in Page 7 Section 4 lines 224 to 266.
18. Page 9 lines 295 to 298: Dielectric constant is not a function of temperature. Why wait until the temperature is stable? What not test it immediately after construction so that you can fix damaged area's immediately
Response: According to the paper “Influence of temperature on the dielectric properties of asphalt mixtures”, the asphalt pavement mixtures will exhibit different dielectric properties under dissimilar temperature conditions. With this in mind, we wait for the asphalt pavement to fully return to normal temperature before carrying out the test to avoid inaccurate prediction results caused by uneven temperature and high temperature.
Special thanks to you for your good comments.

Reviewer 2 Report
The mathematical calculations and experimentally obtained values do not raise doubts. The work is scientifically well thought out, a large amount of material is structured and logical conclusions on the applied problem are given. The language corresponds to a scientific publication. I recommend this paper for consideration in the journal. However, minor remarks require revision.
- GPR scanning consists 0.009 m. This is quite a superficial study of asphalt pavement and destruction can occur not only from the surface, but also due to underground influences, such as groundwater, seismological activity, etc. If you do not consider these factors, it is necessary to specify in the text for which regions and in which climatic conditions the proposed approaches can be applied.
- Figure 4 is not referenced in the text.
Author Response
Dear Editors and Reviewers:
Thank you for your letter and for the reviewers’ comments concerning our manuscript"entitled “Low Compaction Level Detection of Newly Constructed Asphalt Pavement Based on Regional Index"(ID:1891355). Those comments are all valuable and very helpful for revising and improving our paper, as well as the important guiding significance to our researches. We have studied comments carefully and have made correction which we hope meet with approval. Revised portion are marked in red in the paper. The main corrections in the paper and the responds to the reviewer's comments are as flowing:
1.- GPR scanning consists 0.009 m. This is quite a superficial study of asphalt pavement and destruction can occur not only from the surface, but also due to underground influences, such as groundwater, seismological activity, etc. If you do not consider these factors, it is necessary to specify in the text for which regions and in which climatic conditions the proposed approaches can be applied.
Response: I can't find the relevant content of your opinion in the paper. Could you give us the exact location?
2.Figure 4 is not referenced in the text.
Response: We are very sorry for our incorrect writing. All the figure numbers in the paper have been rearranged and rechecked.
Special thanks to you for your good comments.
We tried our best to improve the manuscript and made some changes in the manuscript. These changes will not influence the content and framework of the paper. And here we did not list the changes but marked in red in revised paper.
We appreciate for Editors/Reviewers' warm work earnestly, and hope that the correction will meet with approval.
Once again, thank you very much for your comments and suggestions.